# Massachusetts’ Findings from Statewide Newborn Screening for Spinal Muscular Atrophy

**DOI:** 10.3390/ijns7020026

**Published:** 2021-05-23

**Authors:** Jaime E. Hale, Basil T. Darras, Kathryn J. Swoboda, Elicia Estrella, Jin Yun Helen Chen, Mary-Alice Abbott, Beverly N. Hay, Binod Kumar, Anne M. Counihan, Jacalyn Gerstel-Thompson, Inderneel Sahai, Roger B. Eaton, Anne Marie Comeau

**Affiliations:** 1New England Newborn Screening Program, University of Massachusetts Medical School, Worcester, MA 01605, USA; binod.kumar@umassmed.edu (B.K.); anne.counihan@umassmed.edu (A.M.C.); Jacalyn.Thompson@umassmed.edu (J.G.-T.); inderneel.sahi@umassmed.edu (I.S.); roger.eaton@umassmed.edu (R.B.E.); anne.comeau@umassmed.edu (A.M.C.); 2Department of Neurology, Boston Children’s Hospital, Harvard Medical School, Boston, MA 02115, USA; Basil.Darras@childrens.harvard.edu (B.T.D.); Elicia.Estrella@childrens.harvard.edu (E.E.); 3Center for Genomic Medicine, Department of Neurology, Massachusetts General Hospital, Harvard Medical School, Boston, MA 02114, USA; kswoboda@mgh.harvard.edu (K.J.S.); Jin.Chen@mgh.harvard.edu (J.Y.H.C.); 4Baystate Medical Center, Division of Medical Genetics, Department of Pediatrics, University of Massachusetts Medical School-Baystate, Springfield, MA 01199, USA; MaryAlice.Abbott@baystatehealth.org; 5Division of Genetics, Department of Pediatrics, University of Massachusetts Medical School, Worcester, MA 01605, USA; beverly.hay@umassmemorial.org

**Keywords:** newborn screening, Spinal Muscular Atrophy, *SMN1* gene, *SMN2* gene

## Abstract

Massachusetts began newborn screening (NBS) for Spinal Muscular Atrophy (SMA) following the availability of new treatment options. The New England Newborn Screening Program developed, validated, and implemented a screening algorithm for the detection of SMA-affected infants who show absent *SMN1* Exon 7 by Real-Time™ quantitative PCR (qPCR). We screened 179,467 neonates and identified 9 SMA-affected infants, all of whom were referred to a specialist by day of life 6 (average and median 4 days of life). Another ten *SMN1* hybrids were observed but never referred. The nine referred infants who were confirmed to have SMA were entered into treatment protocols. Early data show that some SMA-affected children have remained asymptomatic and are meeting developmental milestones and some have mild to moderate delays. The Massachusetts experience demonstrates that SMA NBS is feasible, can be implemented on a population basis, and helps engage infants for early treatment to maximize benefit.

## 1. Introduction

Recent advances in the treatment of Spinal Muscular Atrophy (SMA) promoted interest in the development of reliable assays for use in United States (US) newborn screening (NBS) programs. SMA is a progressive neuromuscular disease resulting from a deficiency of the Survival Motor Neuron (SMN) protein that is caused by bi-allelic pathogenic variants in the *SMN1* gene; it is the leading genetic cause of death for infants. Approximately 95% of patients with SMA have a homozygous deletion of Exon 7 of the *SMN1* gene. The clinical presentation of SMA is often modified by the number of copies of a paralog gene, *SMN2*, which differs from *SMN1* by only five distinct base pairs.

Massachusetts and Utah were the first states in the US to offer statewide NBS for SMA. As early as 2015, Massachusetts had established an SMA working group including newborn screeners and clinician specialists to develop guidance for eventual implementation of statewide SMA NBS. Following the successful demonstration of a population-based SMA screening program in Taiwan [1] and the availability of new treatment options targeting the modulation of the *SMN2* gene [2], with other treatments such as gene therapy in the pipeline, the Massachusetts Newborn Screening Advisory Committee voted to offer SMA NBS. The Commonwealth of Massachusetts began offering statewide NBS for SMA in January 2018. In July 2018, the US Secretary of Health and Human Services approved the addition of the disorder, “SMA due to homozygous deletion of Exon 7 in *SMN1*”, to the Recommended Uniform Screening Panel (RUSP) [3]. As of 19 March 2021, 29 states were offering universal screening for SMA, accounting for approximately 60% of US births [4].

## 2. Materials and Methods

### 2.1. Assay Development

The consensus assay of choice for SMA NBS is a Real-Time™ assay to detect the homozygous absence of *SMN1* Exon 7 [5]; carriers are not identified. During the course of Massachusetts’ 2016–2017 planning for assays and screening algorithms, we worked with colleagues at the US Centers for Centers for Disease Control and Prevention (CDC), testing some novel assays they were developing that would enhance the sensitivity and specificity of the Taiwan algorithm. Specifically, these efforts focused on preventing the generation of false-positive screening results from *SMN1* hybrids. Ultimately, we chose two assays, A and B; the possible genotype associations with the results from testing with the two assays is shown in Figure 1 and our incorporation of these two assays into our high-throughput screening algorithms is shown in Figure 2.

Using modifications of two independent CDC-developed Real-Time™ qPCR-based assays [6,7], we developed a multiplex, tiered testing algorithm (Figure 2). 

We had also interfaced our testing algorithm with our Laboratory Information Management System (LIMS) when our CDC colleagues introduced a simpler assay and the use of Locked Nucleic Acids (LNA) to ensure specific amplification from desired *SMN1* sequences. We incorporated the CDC LNA primers and probes into our scheme but maintained our tiered algorithm while more data accrued about the use of LNAs in the high-throughput environment of a screening program. 

Our Tier 1 is a single assay targeting *SMN1* Exon 7 and *RNaseP* (Assay A) with thermoprofiles set to push the assay to its limits in order to ensure that we are more likely to find all babies with homozygous absence of Exon 7. The absence of observed amplification of Exon 7 prompts Tier 2. Our Tier 2 comprises two assays: an additional assay targeting *SMN1* Exon 7, *SMN1* Intron 7, and *RNaseP* (Assay B) as well as a retest of Assay A used in Tier 1. The Tier 2 assays are performed in triplicate, each using an eluate of DNA prepared for Tier 1 and DNA prepared from two new 3mm punches from the same specimen. The Assay B thermoprofiles are less stringent and amplicons are shorter. Assay B often provides an explanation other than poor specimen quality for an apparent failed amplification of Exon 7 in Assay A by identification of *SMN1* hybrids (infants who carry a variant in the region used to prime the amplification of Exon 7 in Assay A). We also developed a Tier 3 Sanger sequencing assay for a short amplicon to confirm the presence or absence of *SMN1* Exon 7 and extended the use of sequencing in order to provide preliminary data on *SMN2* copy numbers for babies being referred to specialists (see accompanying paper).

### 2.2. DNA Preparation for High-Throughput Assay

An aliquot of the DNA prepared for Severe combined immunodeficiency (SCID) NBS [8] is used for the independently performed SMA assay. The SCID and SMA assays remain independent because the laboratory screening services provided to our varied clients include SCID NBS for all clients and we determined that best practice required a single, unmodified SCID NBS for all.

### 2.3. Statewide Screening Implementation

Statewide SMA NBS was initiated in Massachusetts on 27 January 2018. In keeping with our previous early-adopter implementations of NBS for new disorders from which data are collected for evaluation of feasibility, accuracy and clinical outcomes [9,10,11], we used an Institutional Review Board (IRB)-approved verbal consent protocol. Required adjustments to the postnatal documentation of consent by hospitals and other neonatal providers yielded lower rates of consent than observed in our prior statewide implementations (85% vs. 98%).

### 2.4. Referrals and Outcomes Analyses

Figure 3 provides a flow diagram from the point of referral to specialists. In the very beginning of our screening implementation, specialists ordered *SMN2* copy number analyses and the turnaround time for results was 1–2 weeks. Each specialist would then provide a diagnosis and a copy number report to our centralized database at the NBS program. 

Surveillance for potential false negatives was accomplished by intermittent inquiries to the same specialists, to whom infants with clinical presentations also would be referred. In addition, a single clinical-outcomes check on infants who had NBS results showing an *SMN1* hybrid genotype (reported as *SMN1* present) was performed using a telephone inquiry to the infants’ primary care providers (PCP) for the first six of these infants when all were at least six months of age.

## 3. Results

### 3.1. Screening

We have screened 179,467 Massachusetts infants from 27 January 2018 through 31 January 2021 (approximately 36 months of screening) and have identified 9 SMA-affected infants. To date, there have been no infants known to have had a false-negative screening result, yielding a sensitivity of 100% and negative predictive value of 100%. In the first months of screening, we observed one infant with a false-positive SMA NBS result, yielding a specificity of 99.9% and a positive predictive value of 90% (Table 1).

The specimen belonging to the infant with the false-positive SMA NBS result happened to be our first referral and showed absent Exon 7 in both of Assays A and B in Tiers 1 and 2 of our screen. The repeat NBS filter paper specimen obtained concurrently with diagnostic testing by the pediatric neurologist showed the presence of Exon 7 in Tiers 1 and 2, as did diagnostic testing applied to liquid blood sent to a reference laboratory. We first ruled out the possibility of the original screening specimen (showing absent Exon 7) belonging to another infant by confirming that it had been collected on an outpatient basis at a time when no other infant specimens were obtained. We then investigated the reason for the observed absent Exon 7 values, which continued to be observed on multiple retests. Through a series of mixing experiments in which DNA from the false-positive specimen was mixed in equal amount with DNA from specimens showing present Exon 7, we observed Exon 7 to be reduced to no amplification and we observed *RNaseP* Cq values to increase, which suggested to us the original specimen from this infant contained an inhibitor. Our sequencing assay was performed retrospectively on the original and repeat specimens; both showed the presence of a C nucleotide at position 840 of the *SMN1* gene, further confirming the presence of *SMN1*. 

In our experience, reflexing to Tier 2 is relatively infrequent (approximately 0.2% of infants screened) and the result of the Tier 2 assay is typically available on the same day as that of Tier 1 results. All specimens of all 9 SMA-affected infants showed absence of both Exon and Intron 7. Another set of specimens from 10 infants showed the presence of Exon 7 with an absence of Intron 7 (indicating that the infants’ genes are *SMN1* hybrids), explaining the Tier 1 result. Zero specimens showed evidence of the reciprocal, or *SMN2* hybrid. Specimens from the other 294 infants with results that prompted Tier 2 were determined to have a present Exon 7, or In-Range SMA NBS results, based on a combination of replicate data from Tier 2. A subset of the 294 that had continued only to show absent Exon 7 in the more sensitive assay (Assay A) and present Exon 7 in Assay B were sequenced to confirm the Assay B result by the observed presence of a C nucleotide at position c.840 of the *SMN1* gene. In total, of the 314 infants whose specimens prompted Tier 2 testing, 66% (208/314) were in a neonatal intensive care unit (NICU) or special care nursery (SCN).

### 3.2. Outcomes of Infants Who Were Not Referred, Whose Specimens Initially Showed Absence of SMN1 Due to Presence of SMN1 Hybrid Gene

In keeping with expectations from the Taiwan data [1], we characterized the results from specimens that showed an *SMN1* hybrid gene as in range and these infants were not referred. When the first six infants with such results had attained an age of 6 months, we called their PCPs, explained our purpose and inquired about the general health of these infants and whether there was any concern for a neuromuscular disorder and left a phone contact for future questions or concerns. All responses indicated healthy infants and no concerns (Table 2).

### 3.3. Screening Data and Short-Term Clinical Outcomes of the Nine SMA-Affected Infants

Table 3 shows infants’ ages at various points to diagnosis, treatments, and clinical outcomes as well as *SMN2* copy number obtained from diagnostic testing and prenatal testing. The average and median time from date of NBS specimen receipt to report to PCP was 1 day (range 0–3 days) and both the average and median age of the infants at time of NBS report to their PCPs was 4 days (range 3–6 days). The infants had a mean and median age of 9 and 7 days, respectively, for age at first clinic visit to a specialist. The mean and median days of age at first treatment was 36 and 18 days, respectively. All nine SMA infants identified through newborn screening have been treated with nusinersen, Onasemnogene abeparvovec (gene therapy) or both. One infant also received treatment with Risdiplam in addition to nusinersen and Onasemnogene abeparvovec. Four of the nine children are reported to be well and asymptomatic (cases 1, 2, 5, and 7).

## 4. Discussion

Nine infants with SMA were identified by newborn screening and referred to a specialist before the end of their first week of life. Our tiered algorithm functioned well to identify all known affected infants in a timely manner without overwhelming nurseries, specialists, or primary care providers and without causing undue anxiety to families of unaffected infants.

Our choice of screening algorithm was attributable in part to the timing of our assay evaluations and our algorithm’s interface with our LIMS system. Additionally, we chose a conservative approach, not wanting to rely only on the LNA for sequence specificity within the high-throughput environment of NBS until more data accrued about this use. We continue to use independent assays for SCID and SMA. Until SMA screening is universally required in all states for which we provide screening services, we will continue to perform independent assays for SCID and for SMA. In the meantime, data we generate will be useful for methods comparison and harmonization with CDC proficiency testing panels and states that are performing multiplex SCID/SMA assays.

Our choice to provide *SMN2* copy number data was driven by two factors: a need for universal access to such services and a need to improve turnaround time for these results. Turnaround time for diagnostic results including *SMN2* copy number has greatly improved since we began screening in 2018. In addition, original treatment guidelines required that *SMN2* copy number be no greater than 3 [12]. More recent treatment guidelines [13] do not require limits to the copy number of *SMN2* for treatment, but in order to interpret both natural and treatment histories appropriately, a standardized copy number ascertainment is required. One of our infants was almost 6 months old before his first treatment, but he has 4 copies of *SMN2* and was born prior to the 2020 guidelines recommending treatment. Our average age at first treatment for infants with 3 or fewer copies of *SMN2* is 18 days.

At the time we began screening, we knew that if we had used Assay A of Tier 1 alone, we would generate a significant number of false-positive results. Some of these false positives would be due to the *SMN1* hybrids that are identified by Assay B in Tier 2. We found that there is another subset of specimens that are not *SMN1* hybrids (likely with blood collected from a line or using heparinized capillaries) that continue to show absent Exon 7 amplification in our Assay A in Tiers 1 and 2 but show amplification in Assay B, which are reconciled by our Tier 3 sequencing assay. The preponderance of specimens from NICU babies among those specimens prompting Tier 2 does suggest that there might be a PCR inhibitor that would have more implications for Assay A since NICU specimens are more likely to be collected from a line rather than a direct heel stick.

Our observed incidence of *SMN1* hybrids is 1 in 17,947 [95% Confidence Interval (CI): 1/11,080–1/47,202], which is similar to the incidence of 1 in 24,053 reported by Chien et al. [1] in the Taiwanese population. Historically, SMA has been reported to occur in approximately 1 in 10,000 births [14]. To date, our observed incidence of SMA is 1 in 18,957 [95% CI: 1/12,061–1/57,519], which is lower than might be expected from projections of clinical presentation incidence [15] that were observed prior to generalized but non-standardized availability of pre-pregnancy or prenatal SMA testing recommended by American College of Obstetricians and Gynecologists in 2017 [16]. Whether the lower incidence in Massachusetts is due to a limited population sampling, reproductive choices, or a combination of the two remains to be answered.

Of the nine SMA infants identified, one had a prenatal diagnosis, three were born to parents who were both known to be carriers, and one was born to parents with only a single parent identified as a carrier (the other parent had not been tested). The clinical status reported by PCPs at the time of the NBS report was as follows: of the five high-risk infants (prenatal diagnosis or at least one parent known carriers), three were reported to be well, one had poor tone, and one was reported as already being under the care of a specialist. Of the four infants with usual risk (no reported prenatal or parent carrier testing), three were reported as well and one had low or poor tone and a dislocatable hip. One of the usual risk SMA-affected infants was in a SCN for reasons unrelated to overt signs of SMA (gestational age <37 weeks). Interestingly, the clinical status reported by PCPs did not always match the observations of the specialists and underscores the need for immediate clinical evaluation by a trained specialist. The specialists often noted more subtle clinical signs such as tongue fasciculations or axillary slip in infants who were reported to be well by PCPs. 

All of our SMA-affected infants are alive and four of the nine are currently meeting developmental milestones expected for unaffected infants. The other five have experienced mild to moderate delays. There have been no documented harms introduced by treatment. SMA NBS is feasible and allows for the early detection and treatment of affected infants. We are optimistic that SMA NBS will help to maximize the likely benefits of early treatments.

## Figures and Tables

**Figure 1 IJNS-07-00026-f001:**
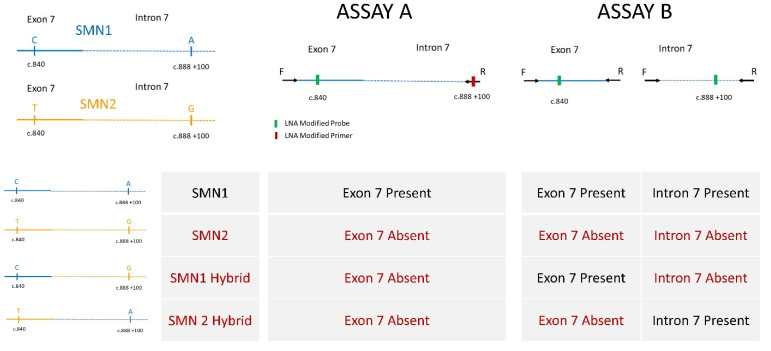
SMN genes and assays to interrogate possible genotypes. Forward and reverse primers and probes shown 5′ to 3′ specific to the amplification of *SMN1* are as follows: (**Assay A** forward: CTTGTGAAACAAAATGCTTTTTAACATCCAT, LNA reverse: ATTGTTTTACATTAACCTTTCAACTTTTT, LNA probe: Cy5/ GGTTTCAGACAA /3IAbRQSp) and (**Assay B** exon: forward CTTGTGAAACAAAATGCTTTTTAACATCCAT, Assay B exon reverse GAATGTGAGCACCTTCCTTCTTTTT, Assay B exon LNA probe Cy5/TTGTCTGAAACC/3IAbRQSp and Assay B intron forward TTGTGGAAAACAAATGTTTTTGAACA, Assay B intron reverse GTAGGGATGTAGATTAACCTTTTATCTAATAGTTT, and Assay B intron LNA probe HEX/CAACTTTTTAACATCT/3IABkFQ).

**Figure 2 IJNS-07-00026-f002:**
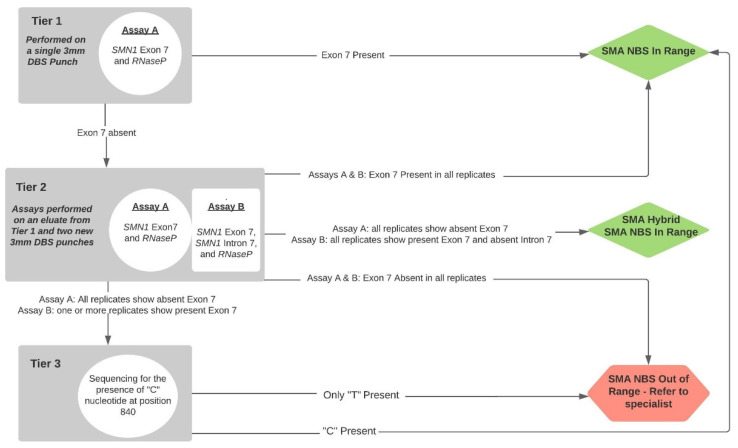
SMA NBS screening laboratory and referral algorithm.

**Figure 3 IJNS-07-00026-f003:**
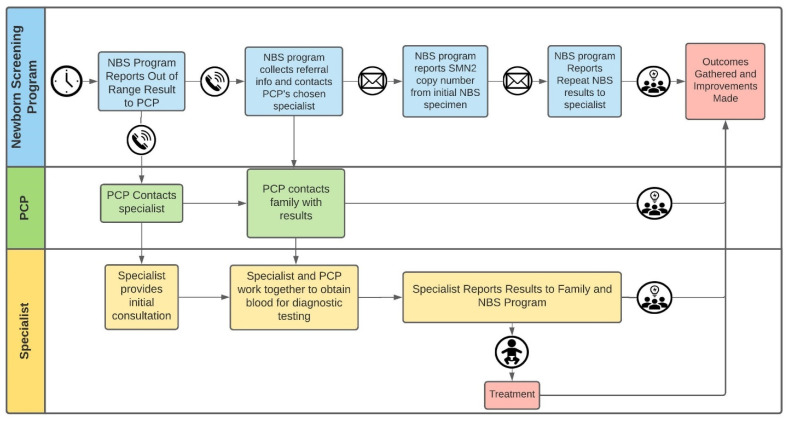
Communications of SMA NBS results and data between NBS program, PCP, and specialist.

**Table 1 IJNS-07-00026-t001:** Findings from the first 36 months of Massachusetts’ SMA NBS.

Infants Whose Specimens	SMA NBS Interpretation	Number of Infants
Showed Exon 7 to be present in Tier 1	In Range	179,153
Showed Exon 7 to be absent in Tier 1		
Showed absent Exon 7 in Tier 2	Out of Range	10 ^1^
Showed evidence of an *SMN1* hybrid in Tier 2	In Range	10
Showed evidence of present *SMN1* in Tier 2	In Range	294
TOTAL SCREENED		179,467

^1^ All were referred to specialists; includes one baby with a false-positive NBS result.

**Table 2 IJNS-07-00026-t002:** Clinical follow up of *SMN1* hybrids.

Age (Months) at Follow up Call	Age (Months) at Last Visit to PCP	Reported Status
17.8	16.0	very healthy, runs around the room, no neuromuscular clinical concerns
14.6	12.0	no neuromuscular clinical concerns; does have unrelated genetic diagnosis
14.0	12.0	well and no neuromuscular clinical concerns
9.1	6.4	no neuromuscular clinical concerns; umbilical hernia
7.7	Not reported	well and no neuromuscular clinical concerns
6.8	3.7	well child visit; normal strength and tone; growth 75th percentile

**Table 3 IJNS-07-00026-t003:** SMA-affected infants identified by NBS.

Case	*SMN2* Copy Number (from Diagnostic Lab)	Age at Specimen Collection/Receipt by NBS/When Screening Result Communicated (Days of Life)	Prenatal Testing	Clinical Status Reported by PCP at Time of NBS Report	Age at First Visit to Specialist (Days of Life)	Clinical Status Reported by Specialist at First Evaluation	Treatment Type	Age at First Treatment (Days of Life)	Current Age (Months)	Current Clinical Status
1	2	2/2/3	Both parents known carriers	Well	26	No symptoms	Nusinersen	38	24	Normal developmental progression for age
2	4	2/4/5		Well; gaining weight but not quite back up to birthweight	5	Tongue fasciculations and mild generalized hypotonia	Onasemnogene abeparvovec	171	22	Continues to develop normally without symptoms or signs of SMA
3	2	2/4/4	Both parents known carriers	Poor tone	6	Hypotonia, decreased movements, head lag and slip through on exam; Respiratory insufficiency, bell shaped chest, paradoxical breathing	Onasemnogene abeparvovec	18	19.5	Moderate motor delays; sits unassisted, rolls over, moderate head control, cannot lift head from a prone position; able to transfer objects hand to hand. Cannot bear full weight on legs, no crawling or walkingAble to chew and swallow, no secretion problems
4	2	1/3/6		Well	9	Day of life (DOL) 7 normal echo and microcephaly per medical records and + axillary slip; DOL 9 tongue fasciculations and impaired swallowing, generalized weakness and hypotonia	Nusinersen/Onasemnogene abeparvovec/Risdiplam	16/29/309	19	At 15 months, crawling, sitting independently, pulls to stand; walks with a walker. Although required g-tube initially, now 100% oral feeder
5	2	1/2/3		Premature but well; no SMA concerns	7	Paradoxical breathing and tongue fasciculations	Nusinersen/Onasemnogene abeparvovec	11/98	13	At 1 years old, continues to acquire age-appropriate motor milestones: rolling, sitting unsupported, crawling
6	2	2/3/6		Extremely dislocatable left hip, low muscle tone	8	No symptoms	Onasemnogene abeparvovec	29	12	Mild motor delays (probably in part due to bracing for hip dislocation)
7	4	1/2/3	Prenatal diagnosis	Status reported as already under the care of a specialist	0	No symptoms	Nusinersen	8	10	Remains asymptomatic at 9 months of age
8	2	2/2/3	One parent known carrier	Well	13	Tongue fasciculations and reduced deep tendon reflexes	Onasemnogene abeparvovec	13	5.5	At 4 months, continues to acquire improved tone and head control in prone and supported sitting positions and recently began eating purees
9	2	2/5/6	Both parents known carriers	Well	7	Limited movements of bilateral lower extremities	Onasemnogene abeparvovec	19	5	Weakness of bilateral lower extremities left > right

## Data Availability

Data beyond what is presented in the article are not publicly available as they are derived from private health information.

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
