# Peer review of "Massachusetts’ Findings from Statewide Newborn Screening for Spinal Muscular Atrophy"

_2409-515X, 2021, doi:10.3390/ijns7020026_

Round 1
Reviewer 1 Report
This is a timely and important study that reports on the state-wide NBS for SMA. The presented data, due to it's nature, is self explanatory and requesting modifications would not be relevant (the report presents the findings of a collection of statewide screening centres, and collectively these data are important and of great interest to the scientific and clinical community). The assay is acknowledged to be as accurate as possible and the data is valid. I have no suggested corrections- it was a paper that I enjoyed reading and I would recommend accepting for publication without correction.
Author Response
Thank you for the review. We are pleased that the study was well accepted and valued.
Reviewer 2 Report
The manuscript written by Hale et al. described their experience of newborn screening for SMA (SMA-NBS) in Massachusetts. They screened 179,467 neonates and identified 9 SMA-affected infants, all of whom were referred to a specialist by day of life 6. Some SMA children identified via SMA-NBS have remained asymptomatic and are meeting developmental milestones and some have mild to moderate delays.
This paper is very nice. I also learned many things from the authors’ experience of SMA-NBS in Massachusetts. But I have several questions. I would like the authors to answer my questions in the revised version of the manuscript.
Major issues
(1) I would like to ask the authors to clarify how the authors identify the false-positive case (Line143), and explain to me more correctly their “inhibitor theory”.
The authors experienced the false-positive case in which SMN1 exon 7 appeared to be absent in both Assay A and Assay B. They explained that the false positives were due to inhibitors.
For the judgement of “SMN1 absence”, it is necessary to confirm that the reference gene was detected in the assay system. Was the reference gene (the RNaseP gene in their assays) detected in this false positive case in Assays A and B? If the reference gene was not detected in these assays, you could not judge as “SMN1 absence”
To explain their judgement “SMN1 absence” in this case, the authors mention the possibility of an “inhibitor”. It follows from their inhibitor theory that the inhibitor may inhibit SMN1 detection specifically. The inhibitor may not inhibit RNaseP detection. It seems very strange.
(2) I would like to know what results were obtained through the mixing experiments with other DNA from DBS specimens showing present Exon 7. I would like the authors to describe this part more clearly.
The authors wrote, “(Lines 156-157) Through a series of mixing experiments with other DNA from DBS specimens showing present Exon 7, we concluded that the original specimen from this infant contained an inhibitor.” The results of the mixing experiments are not shown here.
Minor issues
I would like to know their opinion about independent or simultaneous screening for SMA and SCID.
(1) According to this report, the authors are to continue independent screening for SMA and SCID. The authors appear to predict some benefits of independent screening for SMA and SCID. I would like the authors to explain the benefits of independent screening for SMA and SCID.
(2) The possibility remains that they may anticipate some disadvantages of simultaneous screening of SMA and SCID. If so, I would like to know the disadvantages of simultaneous screening of SMA and SCID.
